# The Effects of 5,6,7,8,3′,4′-Hexamethoxyflavone on Apoptosis of Cultured Human Choriocarcinoma Trophoblast Cells

**DOI:** 10.3390/molecules25040946

**Published:** 2020-02-20

**Authors:** Mengling Zhang, Rui Zhang, Jian Liu, Hongliang Wang, Zhen Wang, Juan Liu, Yang Shan, Huanling Yu

**Affiliations:** 1Longping Branch Graduate School, Hunan University, Changsha 410082, China; zhangmengling1991@163.com (M.Z.); hnliujian@outlook.com (J.L.); wangzhen92@hnu.edu.cn (Z.W.); 2School of Public Health, Beijing Key Laboratory of Environmental Toxicology, Capital Medical University, Beijing 100069, China; 17801084715@163.com; 3Hunan Agriculture Product Processing Institute, Hunan Academy of Agricultural Sciences, Changsha 410125, China; 4Hunan Key Lab of Fruits & Vegetables Storage, Processing, Quality and Safety, Hunan Agricultural Products Processing Institute, Changsha 410125, China; liujmax2019@163.com; 5School of Medical Humanity, Peking University, Beijing 100191, China; zhangrui2500@126.com

**Keywords:** nobiletin, safety, trophoblast, BeWo cells, apoptosis

## Abstract

5,6,7,8,3,4′-Hexamethoxyflavone, also called nobiletin (NOB), widely found in the citrus peel, is one of the main byproducts in citrus processing. NOB is considered safe, but its safety for women during pregnancy is unknown. Therefore, the effect of NOB on apoptosis in human choriocarcinoma trophoblast cells (BeWo cells) was evaluated. Cells were divided into four groups and cultured with different concentrations of NOB (0, 10, 33, and 100 μM) for 12, 24, 36, and 48 h respectively. Cell viability was detected by CCK-8 assay, cell morphology was detected by a Cell Imaging Multi-Mode Reader, and cell cycle and apoptosis were detected by flow cytometry. Cleaved PARP level, the expressions of B cell lymphoma 2 (BCL2) family proteins, and p53 pathway proteins were detected by Western blot. The results showed that after 48 h of cell culture, the cell viability was decreased significantly, but apoptosis was significantly increased. Compared to the cells without NOB treatment, the cells treated with NOB at 10 or 33 μΜ showed no significant differences in the number of suspended cells or late apoptosis rate, except the increase of cell viability. Treatment of NOB at the concentration of 100 μM improved cell viability, attenuated apoptosis, decreased suspended cells, and did not alter the G1 phase arrest, compared with the non-NOB-treated group after 48 h of culturing. The 100 μΜ NOB treatment increased the levels of BCL2 and BCLX_L_, and decreased p53 accumulation in BeWo cells at 48 h, but had no effect on the expression of BAX, BAK, BAD, p21, and G1 phase arrest. These findings provide evidence that NOB (10, 33, and 100 μΜ) was safe for BeWo cells. NOB at the concentration of 100 μΜ could attenuate apoptosis in BeWo cells, which might be helpful to prevent pregnancy-related diseases caused by apoptosis.

## 1. Introduction

Abundantly existing in the peels of *Citrus*, 5,6,7,8,3′,4′-hexamethoxyflavone, also known as nobiletin (NOB), is a flavonoid containing six methoxy groups [1] (Figure 1). Multiple biological activities associated with NOB have been reported, such as antioxidant effects [2,3], anti-inflammatory effects [4,5], antitumor effects [6,7], and cardiovascular-protective properties [8,9]. Several studies have demonstrated that NOB induces apoptosis in various cancer cell lines, including breast cancer MCF-7 cells [10], ovarian cancer SKOV3/TAX cells [11], human gastric cancer SNU-16 cells [12], and BFTC bladder cancer cells [13]. In addition to directly inducing tumor cell apoptosis, NOB has been reported to enhance the pro-apoptotic effect of oxaliplatin on colorectal cancer cells [14]. In non-tumor cells, NOB had an important protective effect on apoptosis induced by external factors. It has been reported that NOB inhibited apoptosis induced by thapsigargin in INS-1D β-cells [15] and hypoxia/reoxygenation in H9c2 cardiomyocytes [16]. After islets transplantation in humans, apoptosis of isolated islets cells could also be ameliorated by NOB [17].

During early pregnancy, trophoblasts precisely differentiate and develop into placenta [18]. Apoptosis is crucial for cellular homeostasis, and plays an important role in many defense mechanisms against infections, mutations, or cellular damage [19]. However, the enhancement or early appearance of apoptosis may lead to pregnancy-related diseases such as preeclampsia, stillbirth, and placental abruption [20,21,22]. Citrus pomace is widely found in the food industry. The FDA approves its addition to orange juice, fruit juice blends, fruit juice drinks, spoon sauces, cereal bars, hot cereals, pastries, etc. [23]. As a unique flavonoid resource in citrus pomace, NOB is generally recognized as safe (GRAS) [23], and is easily accessible to pregnant women. However, there are few reports on the effect of NOB on the apoptosis of human trophoblast cells. 

BeWo cells have been widely used as a model of the villous cytotrophoblast, as they have greater stability, longevity, passage survivability, and absence of patient variability [24,25]. BeWo cells proliferate continually and invade, just like cancerous cells [19], however, the tight regulation of those processes contrasts with those of tumor cells [26]. BeWo cells are thus recognized as a pseudo-tumorigenic tissues [19]. Since NOB is a phytochemical that induces apoptosis in tumor cells, it is rational to hypothesize that NOB might induce apoptosis in BeWo cells.

Therefore, the objective of this study was to determine whether NOB induces apoptosis in human trophoblastic BeWo cells. Cell viability and cell cycle were also measured to analyze the effect of NOB on the viability of BeWo cells. Finally, the expressions of B cell lymphoma 2 (BCL2) family proteins and p53-related proteins were measured to investigate the effect of NOB on apoptosis.

## 2. Results

### 2.1. Cellular Uptake of NOB

BeWo cells were exposed to different doses of NOB (10, 33, and 100 μM), and then cultured for 48 h. The absorption rate of NOB by cells was measured every 12 h. Cellular absorptivities at 12 h were 21.22%, 20.44%, and 23.44% in cells treated with 10, 33, and 100 μM of NOB, respectively (Table 1). The absorptivity of cells treated with 10 and 33 μM of NOB significantly increased to 32.22% and 30.77% respectively after 24 h, and then decreased over time (36 to 48 h). The cells treated with 100 μM of NOB had a decreasing NOB absorption rate of 23.44% at 12 h to 9.17% at 48 h.

### 2.2. The Effect of NOB on the Cell Morphology of BeWo Cells

With the extension of culture time (24, 26, 48 h), the cell proliferation was obvious, and the dead cells, cell debris, and metabolites suspended in the culture medium increased significantly (Figure 2A). The cells that floated in the culture medium were counted by a Cytation™ 5 Cell Imaging Multi-Mode Reader (Figure 2B). The number of suspended cells increased significantly after 48 h of incubation in non-NOB-treated cells. Cells treated with 100 μM of NOB significantly reduced the number of suspended cells compared with non-NOB-treated cells after 36 and 48 h culturing.

### 2.3. The Effect of NOB on the Viability of BeWo Cells

Except in the group treated with NOB 10 µM, the cell viability of BeWo cells in other groups decreased significantly after 48 h culturing, compared with the cell viability at 36 h (Figure 3). The cell viability was increased after exposure to NOB at the concentrations of 10, 33, and 100 μM, compared with non-NOB-treated cells after 48 h of culturing.

### 2.4. The Effect of NOB on Cell Cycle Distribution of BeWo Cells

For non-NOB-treated cells, the number of cells in the G1 phase significantly decreased after 24 h of culturing, and the G1 phase was arrested after 36 and 48 h of culturing (Figure 4). NOB in low concentrations (10 and 33 μM) had no statistically significant effect on cell cycle distribution in sub-G1, G1, S, and G2/M phases of BeWo cells after 48 h of culturing, compared with non-NOB-treated cells. In response to NOB treatment (100 μM), the arrest of the sub-G1 phase and G2/M phase were elevated, and the G1 proportion was not statistically significantly affected.

### 2.5. The Effect of NOB on the Apoptosis of BeWo Cells

Annexin V and PI stainings were used to conduct flow cytometric analysis of apoptotic status (Figure 5). Early apoptosis of non-NOB-treated cells increased after 24 and 36 h of culturing, and then decreased after 48 h of culturing. No significant difference in late apoptosis of non-NOB-treated cells manifested when measured after 12, 24, and 36 h of culturing, while a remarkable increase of late apoptosis after 48 h of culturing was observed. Compared with non-NOB-treated cells, no significant difference in the proportion of apoptotic cells (early, late, total) was revealed following low-dose NOB (10 and 33 μM) treatment in cells. Late apoptosis of cells decreased in response to the 100 μM of NOB treatment compared with non-NOB-treated cells after 48 h of culturing.

Cl-PARP, a marker of caspase-mediated apoptosis, was analyzed by Western blot (Figure 6). Cl-PARP expression of non-NOB-treated cells was augmented over time (24–48 h). The level of cl-PARP was decreased by cells treated with NOB at the concentration of 100 μM, compared with non-NOB-treated cells after 48 h culturing. In contrast, cl-PARP levels were not affected by low dose of NOB (10 or 33 μM).

### 2.6. The Effect of NOB on the Expression of BCL2 Family Proteins in BeWo Cells

BCLX_L_, BAX, BAD, and BAK protein levels were increased; the protein ratio of BCL2/BAX was reduced; and BCL2 expression was not affected in the non-NOB-treated cells which were cultured for 48 h versus cells cultured for 12 h (Figure 7). The levels of BCL2 and BCLX_L_, as well as the ratio of BCL2/BAX, were increased in response to cells treated with 100 μM of NOB, compared with non-NOB-treated cells after 48 h of culturing.

### 2.7. The Effect of NOB on Expression of p53-Related Proteins in BeWo Cells

The expressions of p53 and p21 were significantly increased and the level of mouse double minute 2 homolog (MDM2) was decreased in non-NOB-treated cells cultured for 48 h compared with cells cultured for 12 h (Figure 8). Treatment of the cells with 100 μM of NOB significantly increased MDM2 levels, decreased p53 expression, and had no effect on the levels of p21, compared with non-NOB-treated cells after 48 h of culturing.

## 3. Discussion

NOB is a natural phytochemical with excellent biological effects as a common ingredient in daily nutrition [1]. However, the effects of NOB on the apoptosis of human choriocarcinoma trophoblast cells is not well understood. The present study observed the dosage and time effects of NOB on cell growth and apoptosis in BeWo cells. There was no significant difference in the number of suspended cells or late apoptosis rate between the group treated with low-dose NOB (10 or 33 μM) and non-NOB-treated group. Compared with non-NOB-treated cells, cell viability was increased, and the number of suspended cells and apoptosis was decreased in response to NOB (100 μM) treatment after 48 h of culturing. The treatment of 100 μM NOB after 48 h resulted in the up-regulation of BCL2 and BCLX_L_ and the down-regulation of p53, thereby preventing caspase activation and reducing apoptosis.

There are numerous reports in the literature indicating that NOB had a protective effect on apoptosis induced by environmental factors. NOB was found to prevent the apoptosis of neurons in rats with ischemic brain damage [27,28]. NOB was reported to protect against L-arginine-induced apoptosis in murine pancreatic cells [29]. Meanwhile, NOB has also been reported to promote the apoptosis of tumor cells. Generally speaking, within the experimental dose range of NOB, the apoptosis of tumor cells was enhanced with the increase of NOB concentration. However, some studies observed that the apoptosis of cancer cells was promoted by a low dose of NOB (40 μM) [30], while it was not altered with high doses of NOB (60 or 100 μM) [31]. The pro-apoptotic effect of NOB was also affected by culture time. The apoptosis of MCF-7 human breast cancer cells was enhanced in response to NOB treatment for 24 h [32], but not for 48 or 72 h [31]. The results of the present study found that low doses of NOB (10 and 33 μM) had no significant effects on the apoptosis of BeWo cells, while high doses of NOB (100 μM) alleviated the apoptosis of cells after 48 h of culturing.

Mitochondria play important roles in both intrinsic and extrinsic pathways of apoptosis [33,34]. As a result of apoptotic signals, the BCL2 family is activated, then undergoes a series of physiological and biochemical changes; in turn, the caspase is activated and therefore induces apoptosis [35]. In the present study, the levels of antiapoptotic proteins BCL2 and BCLX_L_ were upregulated in response to BeWo cells treated with NOB (100 μM) for 48 h, which suggested that NOB attenuated apoptosis by blocking a step which was critical for the activation of caspases.

The p53 tumor suppressor gene is a transcription factor that mediates apoptosis by activating the mitochondrial pathway in a hypoxic environment [36]. Meanwhile, after DNA damage, p53 regulates the cell cycle by activating DNA repair proteins [37]. Generally, DNA damage or stress will lead to the increase of the p53 protein level, in turn inducing p21 transcription and causing cell cycle arrest at the G1 phase, so that cells would survive until finishing the damage repair and stress removal [38]. However, there was no change in the p53 signaling pathway in present study; NOB did not change the p21 protein level and block the cell cycle in the G1 phase.

## 4. Materials and Methods

### 4.1. Chemicals and Reagents

NOB (≥95% by HPLC) was obtained from Aladdin Biochemical Technology (Shanghai, China). F12 (Ham) medium and dimethyl sulfoxide (DMSO) were obtained from Boster Biological Technology (Pleasanton, CA, USA). Fetal bovine serum (FBS) was provided by Corning (New York, NY, USA). The CCK-8 kit was purchased from Multi Sciences (Hangzhou, China). A mixture of penicillin and streptomycin, trypsin, Annexin V-FITC/propidium iodide (PI) apoptosis detection kit, cell cycle detection kit, and BCA protein assay kit were obtained from Keygen (Beijing, China). Cold radio immune precipitation assay (RIPA) buffer and phenylmethylsulfonyl fluoride (PMSF) were obtained from Beyotime (Shanghai, China). The antibodies used for the Western blot assay were all purchased from Abcam (Cambridge, UK). The ECL reagent kit was from GE Healthcare (Buckinghamshire, UK). Other chemicals were of analytical grade and used without further purification.

### 4.2. Cell Culture

BeWo cells were purchased from Peking Union Medical College (Beijing, China). The cells were routinely grown in sterile culture flasks at a density of 1 × 10^5^ cells/mL and cultured in F12 (Ham) medium that contained 15% FBS, 100 U/mL penicillin, and 100 U/mL streptomycin in a humidified cell incubator (95% air, 5% CO_2_) at a constant temperature of 37 °C. Cell culture media were changed every three days. Cells grown to 80–90% confluency were trypsinized and subcultured.

### 4.3. Experiment Protocol

NOB was dissolved in DMSO to make stock solutions and added to the cell culture medium to a final concentration of 0.1% DMSO. The same concentration of DMSO (0.1%) alone was added to the control group. BeWo cells were seeded 1 × 10^5^ cells/mL and cultured for 72 h as described above (seen in Section 4.2). After removing the media, the cells were treated with different concentrations of NOB (0, 10, 33, and 100 μM) in serum-starved medium. The cells were collected 12, 24, 36, and 48 h later for subsequent detection, respectively.

### 4.4. Cellular Uptake of NOB

The cells after the experiment protocol were washed with DPBS 3 times. Then, the cells were trypsinized, centrifuged, and harvested in F12 medium. Sonication was used to lyse the cells. The sonication parameters were as follows: the ultrasonic power was 300 W, the ultrasonic time was 5 s/time, the interval was 10 s, and the ultrasonic lysis was 6 times. The intracellular amount of NOB was measured by the standard curve of NOB using an Elx808TM microplate reader (Biotek, Winooski, VT, USA). The standard curve of NOB absorption at 334 nm was obtained using NOB standards (≥95% by HPLC). The standard curve was y = 0.002x + 0.14, R^2^ = 0.985, where x was the concentration of NOB (μM) and y was the absorbance. Then, the cellular uptake (CU) of NOB was calculated as follows:(1)$$\mathrm{CU}=\frac{A}{B}\times 100\%$$
where CU was the cellular uptake of NOB, A was the intracellular amount of NOB, and B was the intracellular and extracellular amount of NOB.

### 4.5. Cell Morphological Observation

BeWo cells were seeded in 12-well plates at a density of 1 × 10^5^/mL and cultured for 72 h. After removing the media, the cells were cultured in serum-starved medium supplied with 0, 10, 33, and 100 μM of NOB, respectively. Cell morphology was monitored using a Cytation™ 5 Cell Imaging Multi-Mode Reader (Biotek, Winooski, VT, USA) at the time points of 12, 24, 36, and 48 h. Suspension cells were counted separately using a Cytation™ 5 Cell Imaging Multi-Mode Reader with corresponding Gen5 Image software version 3.08.

### 4.6. Cell Viability Assay

BeWo cell viability was detected using CCK-8 assay according to the manufacturer’s recommendations. BeWo cells were plated into 96-well plates and incubated for 72 h. After the experiment protocol (seen in Section 4.3), cells were treated with CCK-8 solution at 10 μL/well of 96-well plates. After incubation for 4 h at 37 °C, the absorbance at 450 nm was measured using a microplate reader (Biotek, Winooski, VT, USA).

### 4.7. Cell Cycle Assay

The cell cycle detection kit was used to analyze cell cycle distribution, according to the manufacturer’s specifications. After experiment protocol (seen in Section 4.3), cells were trypsinized, fixed with 70% ethanol, and stored at 4 °C overnight. The cells were washed with ice-cold PBS and incubated in PI staining buffer in darkness at room temperature for 30 min. Data acquisition was performed by a flow cytometer (ACEA, San Diego, CA, USA), and cell distribution was analyzed to determine the fractions of cells in the sub-G1, G1, S, and G2/M phases.

### 4.8. Cell Apoptosis Assay

An annexin V-FITC/PI apoptosis detection kit was used to quantify apoptotic cells following the manufacturer’s instructions. After the experiment protocol (seen in Section 4.3), the cells were collected and the cell concentration was adjusted according to the manufacturer’s protocol. The cells were suspended in 500 μL of binding buffer containing annexin V and PI and incubated for 15 min at room temperature. Then, the cell apoptosis was evaluated by using a flow cytometer (ACEA, Chicago, IL, USA). The excitation wavelength was 488 nm, while the detection wavelengths of FITC and PI were set at 525 and 620 nm, respectively. Early apoptotic cells were identified as annexin V-positive/PI-negative cells, and late apoptotic cells were identified as annexin V-positive/PI-positive cells.

### 4.9. Western Blot Analysis

Total proteins in the BeWo cells were lysed with a RIPA buffer containing 1% PMSF. The proteins were quantified using a BCA protein assay kit. A total of 20 μg of proteins were electrophoresed in 10% or 12% sodium dodecyl sulfate polyacrylamide gel electrophoresis (SDS-PAGE), and then the separated proteins were transferred to nitrocellulose membranes. After blocking and washing, the membranes were incubated with primary antibodies at 4 °C overnight with continuous agitation. The working concentrations of primary anti-ACTB and anti-BAD were 1:2000. The working concentrations of primary anti-cleaved PARP, anti-p21, anti-MDM2, anti-BCL2, anti-BCLX_L_, and anti-BAX were 1:1000. The working concentration of primary anti-BAK was 1:10,000. The working concentration of primary anti-p53 was 5 μg/mL. Then, the membranes were incubated with HRP-conjugated secondary antibodies, and proteins were evaluated by an ECL reagent kit. Image J software was employed to calculate the density of protein bands. ACTB was selected as the internal standard.

### 4.10. Statistical Analysis

All experiments were performed in triplicate, and the data were summarized as the mean ± SD. Comparisons were made with a one-way ANOVA, followed by a Duncan’s test. A *p* < 0.05 was considered statistically significant.

## 5. Conclusions

In summary, NOB at 10 and 33 μM had no effect on BeWo cell apoptosis. BeWo cells treated with NOB at the concentration of 100 μM remarkably relieved the apoptosis of cells after 48 h of culturing. These findings suggested that NOB (10, 33, and 100 μM) was safe for BeWo cells, and NOB concentration of 100 μM could delay cell aging and prevent against pregnancy-induced diseases caused by apoptosis.

## Figures and Tables

**Figure 1 molecules-25-00946-f001:**
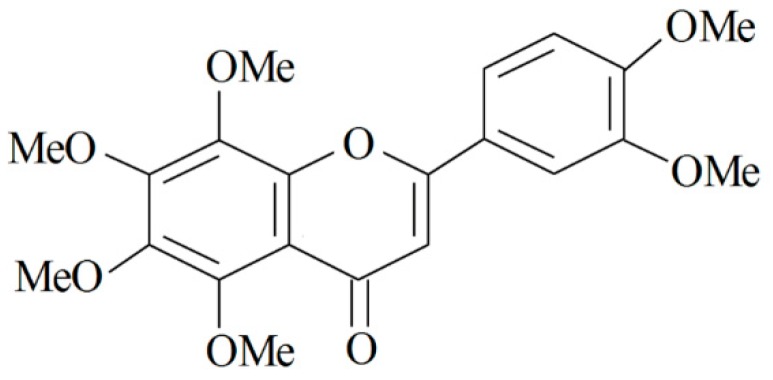
Chemical structure of 5,6,7,8,3′,4′-hexamethoxyflavone (nobiletin, NOB).

**Figure 2 molecules-25-00946-f002:**
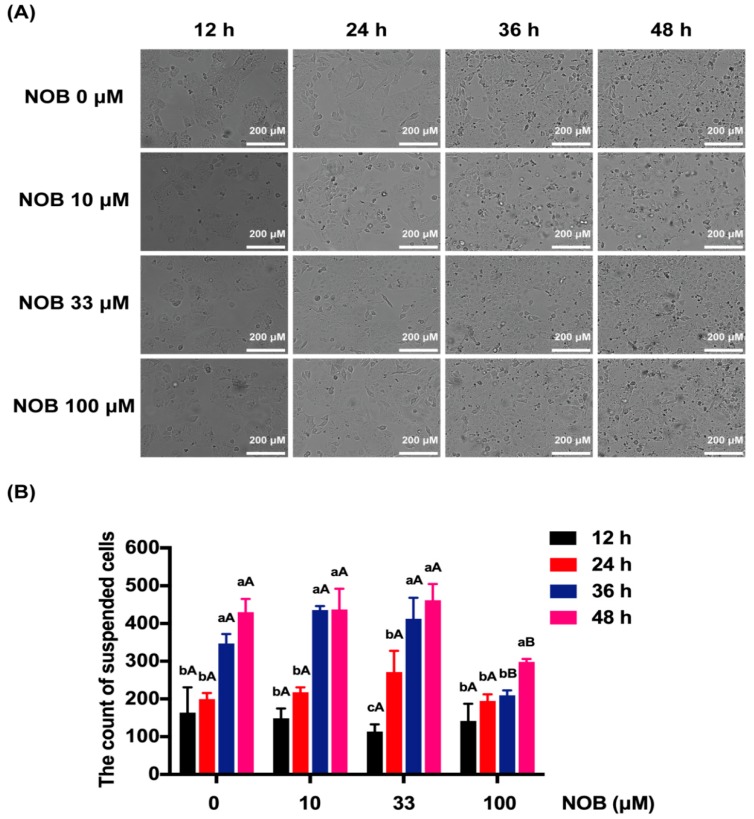
The effect of NOB on cell morphology. (**A**) The morphology of BeWo cells. (**B**) The count of suspended cells. Data were summarized as mean ± SD, n = three independent experiments. At the same NOB treatment dose, different lowercase letters represent significant differences at different treatment times, *p* < 0.05. At the same treatment time, different capital letters represent significant differences at different NOB doses, *p* < 0.05, one-way ANOVA, with Duncan’s multiple range test.

**Figure 3 molecules-25-00946-f003:**
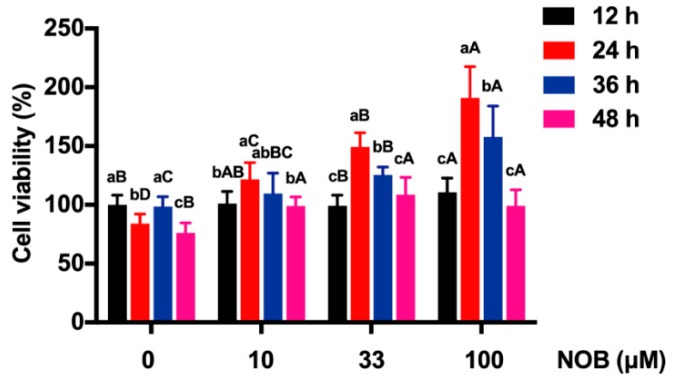
The effect of NOB on the viability of BeWo cells. Data were summarized as mean ± SD, n = three independent experiments. At the same NOB treatment dose, different lowercase letters represent significant differences at different treatment times, *p* < 0.05. At the same treatment time, different capital letters represent significant differences at different NOB doses, *p* < 0.05, one-way ANOVA, with Duncan’s multiple range test.

**Figure 4 molecules-25-00946-f004:**
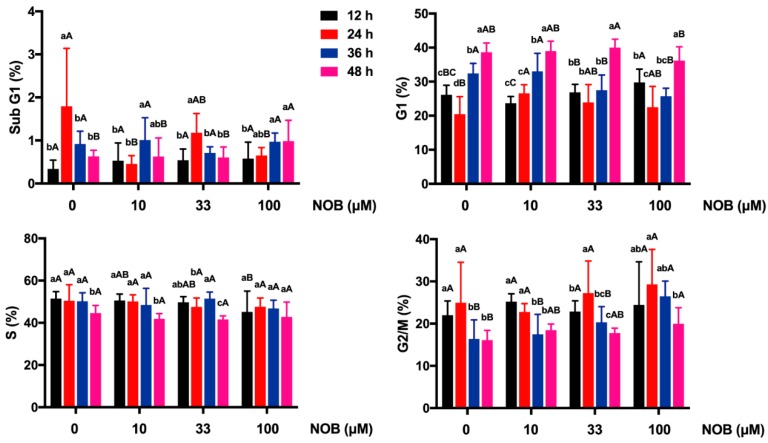
The effect of NOB on cell cycle distribution of BeWo cells. Cell cycle distribution of BeWo cells in subG1, G1, S, and G2/M phases were detected by flow cytometry. Data were summarized as mean ± SD, n = three independent experiments. At the same NOB treatment dose, different lowercase letters represent significant differences at different treatment times, *p* < 0.05. At the same treatment time, different capital letters represent significant differences at different NOB doses, *p* < 0.05, one-way ANOVA, with Duncan’s multiple range test.

**Figure 5 molecules-25-00946-f005:**
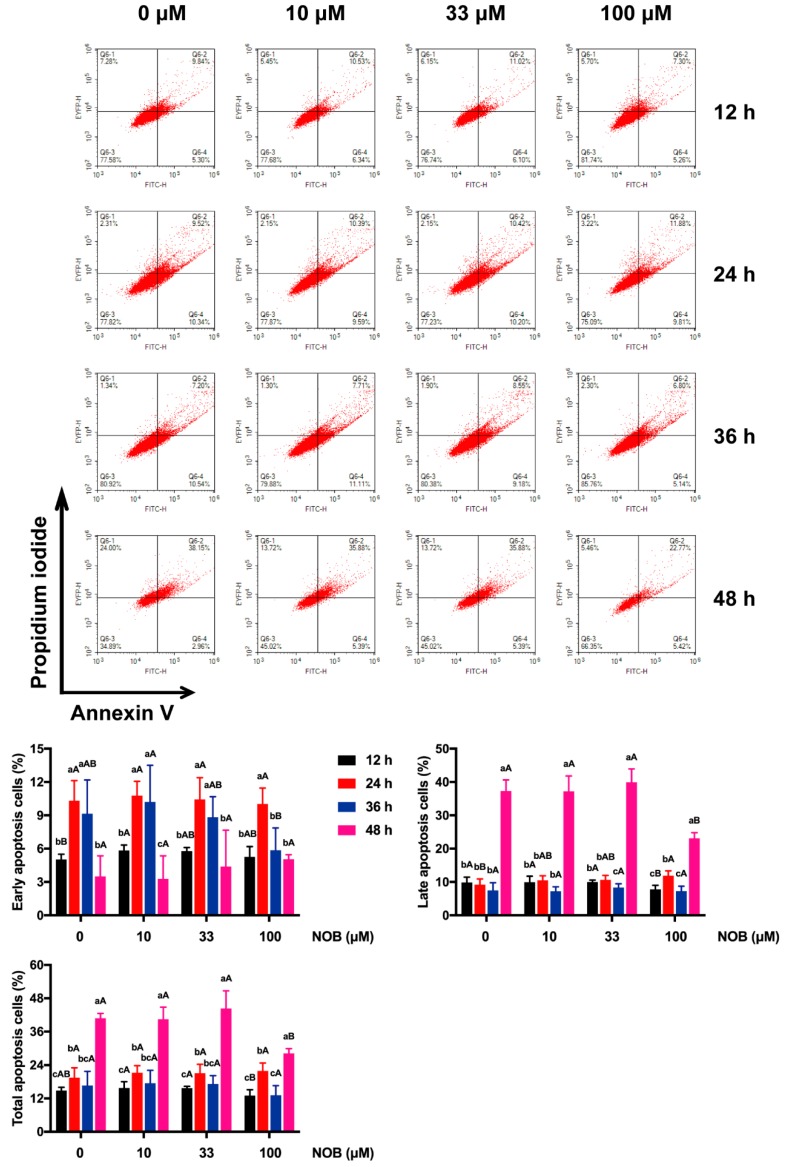
The effect of NOB on apoptosis of BeWo cells. Annexin V and PI fluorescence values were estimated by flow cytometry. Data were summarized as mean ± SD, n = three independent experiments. At the same NOB treatment dose, different lowercase letters represent significant differences at different treatment times, *p* < 0.05. At the same treatment time, different capital letters represent significant differences at different NOB doses, *p* < 0.05, one-way ANOVA, with Duncan’s multiple range test.

**Figure 6 molecules-25-00946-f006:**
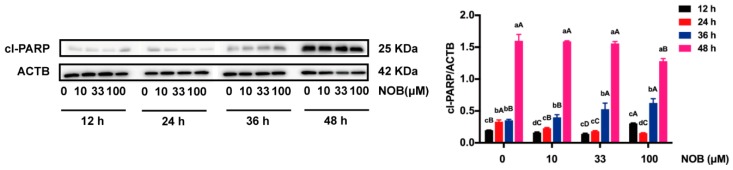
The effect of NOB on cleaved PARP (cl-PARP) of BeWo cells. The expression of cl-PARP was measured by Western blot. Data were summarized as mean ± SD, n = three independent experiments. At the same NOB treatment dose, different lowercase letters represent significant differences at different treatment times, *p* < 0.05. At the same treatment time, different capital letters represent significant differences at different NOB doses, *p* < 0.05, one-way ANOVA, with Duncan’s multiple range test.

**Figure 7 molecules-25-00946-f007:**
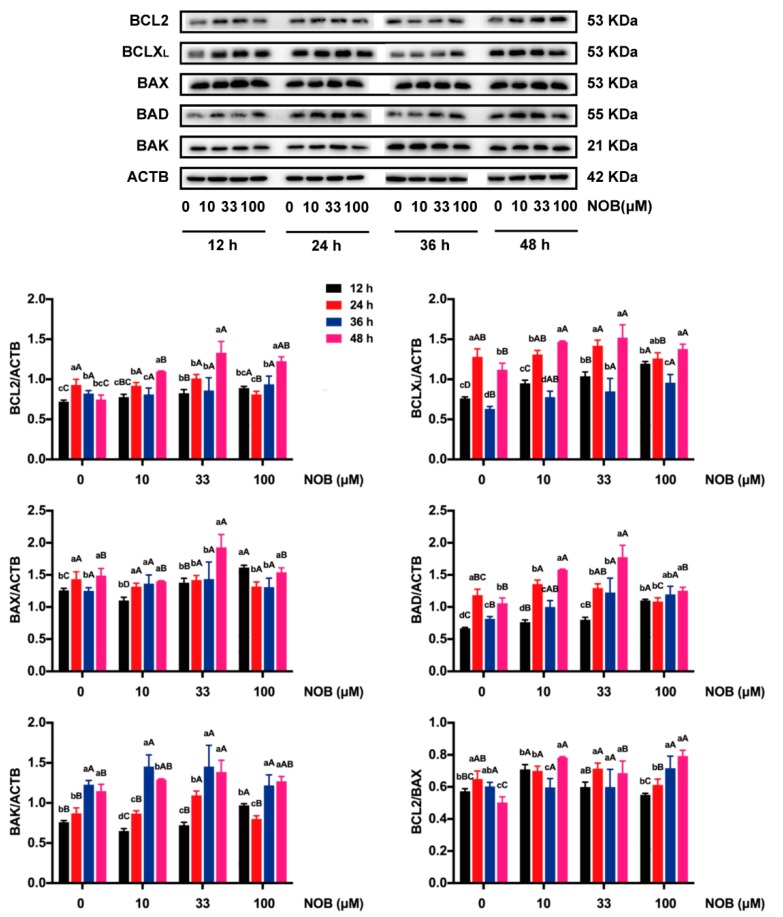
The effect of NOB on expressions of selected members of the B cell lymphoma 2 (BCL2) family genes of BeWo cells. B cell lymphoma 2 (BCL2) family genes in this study included BCL2, BCLX_L_, BAX, BAD, and BAK. The ratio of BCL2/BAX was calculated. The protein expressions of selected members of the BCL2 family genes were estimated by Western blot. ACTB was used as an internal control, and the relative protein expression was the ratio of target protein to ACTB. Data were summarized as mean ± SD, n = three independent experiments. At the same NOB treatment dose, different lowercase letters represent significant differences at different treatment times, *p* < 0.05. At the same treatment time, different capital letters represent significant differences at different NOB doses, *p* < 0.05, one-way ANOVA, using Duncan’s multiple range test.

**Figure 8 molecules-25-00946-f008:**
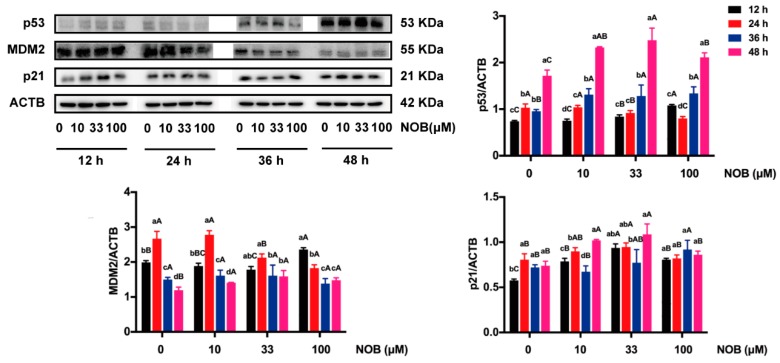
The effect of NOB on expressions of p53 pathway genes of BeWo cells. The protein expressions of p53, MDM2, and p21 were estimated by Western blot. ACTB was used as an internal control, and the relative protein expression was the ratio of target protein to ACTB. Data were summarized as mean ± SD, n = three independent experiments. At the same NOB treatment dose, different lowercase letters represent significant differences at different treatment times, *p* < 0.05. At the same treatment time, different capital letters represent significant differences at different NOB doses, *p* < 0.05, one-way ANOVA, using Duncan’s multiple range test.

**Table 1 molecules-25-00946-t001:** Cellular uptake of NOB.

	12 h	24 h	36 h	48 h
NOB 10 μM	21.22% ± 8.42% ^bA^	32.22% ± 6.67% ^aA^	19.44% ± 9.17% ^bA^	15.56% ± 8.46% ^bA^
NOB 33 μM	20.44% ± 7.17% ^bA^	30.77% ± 9.07% ^aA^	12.12% ± 7.80% ^cB^	17.34% ± 5.31% ^bcA^
NOB 100 μM	23.44% ± 4.20% ^aA^	15.33% ± 3.00% ^bB^	10.17% ± 2.93% ^cB^	9.17% ± 3.08% ^cB^

Data are summarized as mean ± SD, n = three independent experiments. At the same NOB treatment dose, different lowercase letters represent significant differences at different treatment times, *p* < 0.05. At the same treatment time, different capital letters represent significant differences at different NOB doses, *p* < 0.05, one-way analysis of variance (ANOVA), using Duncan’s multiple range test.

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
