# Peer review of "The Effects of 5,6,7,8,3′,4′-Hexamethoxyflavone on Apoptosis of Cultured Human Choriocarcinoma Trophoblast Cells"

_molecules, 2020, doi:10.3390/molecules25040946_

Round 1

Reviewer 1 Report

 In presented study the authors evaluated the effect of NOB ( 5,6,7,8,3',4'-hexamethoxyflavone, also called nobiletin ) on apoptosis in BeWo cells. The work is interesting and important from the point of view of preventing diseases that can develop during pregnancy in women. Nevertheless, I have some slight comments and suggestions.

Line 90  except NOB 33 µM which increased.

Fig 5  Is in poor resolution, I would suggest improving the quality of this figure.

Line 208  How long sonication was used ? Why  nobiletin concentration was not calculated from the standard curve ? Pleas give a parameters of spectrophotometer.
I apologize in advance for maybe a trivial question but I wonder because the significant inhibition of apoptosis after 48 hours at a dose of 100 µM is negatively correlated with the level of NOD uptake, and after that time decreased from 23.44% to 9.17%. How can you explain this relationship?

Author Response

Thanks for your careful read and thoughtful suggests on manuscript. The questions have been answered point-to-point.

Question one: Line 90 except NOB 33 µM which increased.

Answer:

    Thank you for your suggestion. The errors in the manuscript have been corrected as follows.

Before revision:

    Regardless of NOB treatment or not, the cell viability of BeWo cells was decreased significantly after 48 h culturing (Fig. 3).

After revision:

    Except the group treated with NOB 10 µM, the cell viability of BeWo cells of other groups decreased significantly after 48 h culture compared with that at the time point 36 h (Fig. 3).

    The revisions were highlighted with red in the manuscript, line from102 to 103, page 4.

Question two: Fig 5 Is in poor resolution, I would suggest improving the quality of this figure.

Answer:

    Thank you for your suggestion. Figure 5 has been resubmitted in the hope that its quality would be better.

Question three: Line 208 How long sonication was used?

Answer:

     Thank you for your question. The sonication parameters were as follows: the ultrasonic power was 300 W, the ultrasonic time was 5 s /time, the interval was 10 s, and the ultrasonic lysis was 6 times.

     The revisions were highlighted with red in the manuscript, line from 250 to 252, page 11.

Question four: Why nobiletin concentration was not calculated from the standard curve? Please give a parameters of spectrophotometer.

Answer:

Thank you for your suggestion. It was particularly sorry that there was no detailed explanation on the calculation method of NOB cellular uptake. The standard curve of NOB was used to calculate the intracellular concentration of NOB. The standard curve of NOB absorption at 334 nm was obtained using NOB standards (≥ 95% by HPLC). The standard curve is y=0.002x+0.14, R2=0.985. X was the concentration of NOB (μM) and y was the absorbance. Then, The cellular uptake of NOB (CU) was calculated as following.

CU=A/B×100%

where CU was the cellular uptake of NOB, A was the intracellular amount of NOB, and B was the intracellular and extracellular amount of NOB.

The spectrophotometer used was an ELx808TM microplate reader (Biotek, VT, USA). The absorption wavelength was 334 nm.

The revisions were highlighted with red in the manuscript, line from 252 to 256, page from 11 to 12.

Question five: I apologize in advance for maybe a trivial question but I wonder because the significant inhibition of apoptosis after 48 hours at a dose of 100 µM is negatively correlated with the level of NOB uptake, and after that time decreased from 23.44% to 9.17%. How can you explain this relationship?

Answer:

     Thank you for your question.

     First, the cellular uptake of NOB was calculated as the cellular uptake of NOB =A/B×100%. A was the intracellular amount of NOB, and B was the intracellular and extracellular amount of NOB. The higher cell uptake rate of NOB does not mean the higher intracellular amount of NOB. There was a reasonable explanation to the fact that the NOB absorption rate of cells in the high-dose NOB group was lower than that of cells in the low-NOB group.

     Second, after the cells were treated with 100 μM NOB for 48 h, the NOB absorption rate of the cells decreased from 23.44% to 9.17%, compared to culturing for 12 h, which may be due to that the cell's metabolism and excretion of NOB is higher than the cell's uptake of NOB.

     Third, for non-NOB-treated cells, the degree of apoptosis was very low (less than 15% of total apoptosis) within 12-36 h, but after 48-hour culture, the apoptosis reached 40%. NOB had no effect on the normal apoptosis (low level of apoptosis) of cells, but reduced apoptosis when the cells were undergoing abnormal apoptosis. It can be seen that the effect of NOB on inhibiting apoptosis was not only related to the level of NOB uptake, but also depends on environmental factors such as the state of the cells and the culture time.

Reviewer 2 Report

Comments to Author:

In the manuscript molecules-711013 authors explained effects of 5,6,7,8,3',4'-hexamethoxyflavone on apoptosis of cultured human choriocarcinoma trophoblast cells. The method has been explained properly for the adsorbent experiments. However, I have a few concerns that need to be addressed in the manuscript.  

Comments:

Morphological observation on cultured human choriocarcinoma trophoblast cells need to be added in the manuscript. The author didn’t study the DNA fragmentation analysis and MTT assays. I would suggest authors to add those study. Why author have chosen all the designed assays? Is there any specific reason for each of those assays? Author need to present the monitored cell morphology at the time point of 12 h, 24 h, 36 h and 48 h.   Author didn’t show the image for density of protein bands in western blot analysis

Author Response

Thanks for your careful read and thoughtful suggests on manuscript. The questions have been answered point-to-point.

Reviewer 2

Question one: Morphological observation on cultured human choriocarcinoma trophoblast cells need to be added in the manuscript. Author need to present the monitored cell morphology at the time point of 12 h, 24 h, 36 h and 48 h.

Answer:

     Thank you for your suggestion. The morphology of the cells have been characterized in Figure 2A. The manuscript have been corrected as follows.

Before revision:

     The proliferation of BeWo cells was observed over time (24, 36 and 48 h), and the cells floated in the culture medium were counted by a Cytation™ 5 Cell Imaging Multi-Mode Reader (Fig. 2). The number of suspended cells increased significantly after 48 h of incubation in non-NOB-treated cells. Cells treated with 100 μM NOB significantly reduced the number of suspended cells compared with non-NOB-treated cells after 36 and 48 h of culture.

After revision:

     With the extension of culture time (24, 26, 48 h), the cell proliferation was obvious, and the dead cells, cell debris and metabolites suspended in the culture medium increased significantly. The cells floated in the culture medium were counted by a Cytation™ 5 Cell Imaging Multi-Mode Reader (Fig. 2). The number of suspended cells increased significantly after 48 h of incubation in non-NOB-treated cells. Cells treated with 100 μM NOB significantly reduced the number of suspended cells compared with non-NOB-treated cells after 36 and 48 h of culture.

    The revisions were highlighted with red in the manuscript, line from 88 to 89, page 3.

Question two: The author didn’t study the DNA fragmentation analysis and MTT assays. I would suggest authors to add those study.

Answer:

    Thanks for your suggestion. The DNA fragmentation analysis and MTT would more better to show the cell apoptosis and viability, respectively.

    There are two aspects to the physiological changes of apoptosis. On the one hand, activating DNase to cleave chromatin DNA leads to the break of nuclear DNA and promotes the formation of chromatin bodies (Fischer, 2017). On the other hand, activating caspase hydrolyzes proteins and promotes the formation of apoptotic bodies by degrading proteolytic enzymes and inactivating apoptosis inhibitors (Radha & Raghavan, 2017). PARP is a DNA repair enzyme and also the main target of caspase 3 (the most important terminal cleavage enzyme in the process of apoptosis )(J. Chen et al., 2015). After PARP is sheared, it will lose its enzyme activity and reduce its ability to repair DNA (B. S. Chen, Longtine, & Nelson, 2013). Therefore, cl-PARP can also indirectly characterize the degree of DNA fragmentation. The expression of cl-PARP was tested in this paper.

     Both MTT (Tan et al., 2019) and CCK8 (Li et al., 2019) tests are indicators of cell viability. In this paper, cell viability was tested using CCK 8 assay.

Question three: Why author have chosen all the designed assays? Is there any specific reason for each of those assays?

Answer:

    Thanks for your suggestion. It was very sorry that the logical relationship between the indicators was not clear in the manuscript.

    First, cell uptake of NOB, cell viability, and cell morphology were basic indicators in this paper.

    Second, the morphological changes of cell apoptosis were characterized by Annexin V-FITC/PI assay, due to the externalization of phosphatidylserine (PS) in the membrane of early apoptotic cells (Y. S. Chen et al., 2019; Yang, Lim, Bazer, & Song, 2017), and the loss of cell membrane integrity in late apoptotic cells (Liu, Meng, Shi, & Xu, 2018; Panic et al., 2019).

    Third, The physiological changes of cell apoptosis were characterized by cl-PARP. DNA fragmentation and caspase activation are important physiological changes in apoptosis (seen in response of answer two) (Radha et al., 2017). PARP is a DNA repair enzyme and also the main enzyme-cut substrate of caspase 3 (Fischer, 2017). When PARP is severely sheared, it indicates that the DNA is severely broken, and caspase 3 activates and executes apoptosis, which eventually leads to the elimination of apoptotic cells (Siddiqui, Ahad, & Ahsan, 2015).

    Fourth, the p53 signaling pathway (including p53, p21, MDM2) is an important signaling pathway that regulates cell cycle and apoptosis. MDM2 is an E3 ubiquitin ligase involved in p53 degradation (Eischen & Lozano, 2014). MDM2 forms a feedback loop for p53, in which p53 activates MDM2 gene expression, increases MDM2 protein levels, and then MDM2 binds p53 to inhibit p53 activity (Fischer, 2017). Cell cycle arrest is the first defense line mediated by p53. P21, a downstream gene of p53, is required for p53-mediated G1 and G2 cell cycle arrest, of which p21 is more effective in preventing G1 progression (Senturk & Manfredi, 2013). The second defense line mediated by p53 is apoptosis. If hypoxia-induced DNA accumulation damage is severe or unrepairable, the failure of cells to exit the cell cycle from incomplete mitosis may lead to induction of apoptosis (Mrakovcic, Kleinheinz, & Frohlich, 2019).

    Last, BCL2 family genes (including BCL2, BCLXL, BAK, BAX and BAD) are also important signaling pathways that regulate apoptosis. Mitochondria are crucial multifunctional organelles that actively regulate the stability of the intracellular environment and are directly involved in the different and complex interrelated processes that regulate cell survival or death (Apostolova, Blas-Garcia, & Esplugues, 2011; B. S. Chen et al., 2013). The fate of cells apoptosis is usually determined by the balance between anti-apoptotic (BCL2, BCLXL) and pro-apoptotic members (BAK, BAX and BAD) of BCL-2 family through mitochondrial pathway (Luo, Caniggia, & Post, 2014; Shimizu, Eguchi, Kosaka, Kamiike, Matsuda, & Tsujimoto, 1995; Shroff, Snyder, & Chandel, 2007). The lower the BCL2/BAX ratio was meant the higher the apoptotic degree (Perez-Perez et al., 2019).

The logical relationship between these indicators have been indicated where appropriate in the text.

Question four: Author didn’t show the image for density of protein bands in western blot analysis.

Answer:

    Thank you for your suggestion. Thank you for your suggestion. It was particularly sorry that the figures of protein bands and density were separated. Figures 6, 7, and 8 have been rearranged, and updated legends.

References

Apostolova, N., Blas-Garcia, A., & Esplugues, J. V. (2011). Mitochondria Sentencing About Cellular Life and Death: A Matter of Oxidative Stress. Current Pharmaceutical Design, 17(36), 4047-4060. https://doi.org/Doi 10.2174/138161211798764924.

Chen, B. S., Longtine, M. S., & Nelson, D. M. (2013). Punicalagin, a polyphenol in pomegranate juice, downregulates p53 and attenuates hypoxia-induced apoptosis in cultured human placental syncytiotrophoblasts. American Journal of Physiology-Endocrinology and Metabolism, 305(10), E1274-E1280. https://doi.org/10.1152/ajpendo.00218.2013.

Chen, J., Chen, A. Y., Huang, H., Ye, X., Rollyson, W. D., Perry, H. E., . . . Chen, Y. C. (2015). The flavonoid nobiletin inhibits tumor growth and angiogenesis of ovarian cancers via the Akt pathway. Int J Oncol, 46(6), 2629-2638. https://doi.org/10.3892/ijo.2015.2946.

Chen, Y. S., Liu, J. W., Geng, S., Liu, Y. L., Ma, H. J., Zheng, J., . . . Liang, G. Z. (2019). Lipase-catalyzed synthesis mechanism of tri-acetylated phloridzin and its antiproliferative activity against HepG2 cancer cells. Food Chemistry, 277, 186-194. https://doi.org/10.1016/j.foodchem.2018.10.111.

Eischen, C. M., & Lozano, G. (2014). The Mdm Network and Its Regulation of p53 Activities: A Rheostat of Cancer Risk. Human Mutation, 35(6), 728-737. https://doi.org/10.1002/humu.22524.

Fischer, M. (2017). Census and evaluation of p53 target genes. Oncogene, 36(28), 3943-3956. https://doi.org/10.1038/onc.2016.502.

Li, H. Y., Yang, H. G., Li, P., Wang, Y. Z., Huang, G. X., Xing, L., . . . Zheng, N. (2019). Effect of Heat Treatment on the Antitumor Activity of Lactoferrin in Human Colon Tumor (HT29) Model. J Agric Food Chem, 67(1), 140-147. https://doi.org/10.1021/acs.jafc.8b05131.

Liu, R. H., Meng, Q., Shi, Y. P., & Xu, H. S. (2018). Regulatory role of microRNA-320a in the proliferation, migration, invasion, and apoptosis of trophoblasts and endothelial cells by targeting estrogen-related receptor gamma. J Cell Physiol, 234(1), 682-691. https://doi.org/10.1002/jcp.26842.

Luo, D. C., Caniggia, I., & Post, M. (2014). Hypoxia-inducible regulation of placental BOK expression. Biochemical Journal, 461, 391-402. https://doi.org/10.1042/Bj20140066.

Mrakovcic, M., Kleinheinz, J., & Frohlich, L. F. (2019). p53 at the Crossroads between Different Types of HDAC Inhibitor-Mediated Cancer Cell Death. Int J Mol Sci, 20(10). https://doi.org/10.3390/ijms20102415.

Panic, M., Stojkovic, M. R., Kraljic, K., Skevin, D., Redovnikovic, I. R., Srcek, V. G., & Radosevic, K. (2019). Ready-to-use green polyphenolic extracts from food by-products. Food Chemistry, 283, 628-636. https://doi.org/10.1016/j.foodchem.2019.01.061.

Perez-Perez, A., Toro, A., Vilarino-Garcia, T., Guadix, P., Maymo, J., Duenas, J. L., . . . Sanchez-Margalet, V. (2019). Leptin protects placental cells from apoptosis induced by acidic stress. Cell Tissue Res, 375(3), 733-742. https://doi.org/10.1007/s00441-018-2940-9.

Radha, G., & Raghavan, S. C. (2017). BCL2: A promising cancer therapeutic target. Biochim Biophys Acta Rev Cancer, 1868(1), 309-314. https://doi.org/10.1016/j.bbcan.2017.06.004.

Senturk, E., & Manfredi, J. J. (2013). p53 and cell cycle effects after DNA damage. Methods Mol Biol, 962, 49-61. https://doi.org/10.1007/978-1-62703-236-0_4.

Shimizu, S., Eguchi, Y., Kosaka, H., Kamiike, W., Matsuda, H., & Tsujimoto, Y. (1995). Prevention of Hypoxia-Induced Cell-Death by Bcl-2 and Bcl-Xl. Nature, 374(6525), 811-813. https://doi.org/DOI 10.1038/374811a0.

Shroff, E. H., Snyder, C., & Chandel, N. S. (2007). Bcl-2 family members regulate anoxia-induced cell death. Antioxidants & Redox Signaling, 9(9), 1405-1409. https://doi.org/10.1089/ars.2007.1731.

Siddiqui, W. A., Ahad, A., & Ahsan, H. (2015). The mystery of BCL2 family: Bcl-2 proteins and apoptosis: an update. Arch Toxicol, 89(3), 289-317. https://doi.org/10.1007/s00204-014-1448-7.

Tan, Q., Peng, L., Huang, Y., Huang, W., Bai, W., Shi, L., . . . Chen, T. (2019). Structure-Activity Relationship Analysis on Antioxidant and Anticancer Actions of Theaflavins on Human Colon Cancer Cells. J Agric Food Chem, 67(1), 159-170. https://doi.org/10.1021/acs.jafc.8b05369.

Yang, C., Lim, W., Bazer, F. W., & Song, G. (2017). Myricetin suppresses invasion and promotes cell death in human placental choriocarcinoma cells through induction of oxidative stress. Cancer Letters, 399, 10-19. https://doi.org/10.1016/j.canlet.2017.04.014.

Reviewer 3 Report

The paper is very interesting and important because it concerns the safety of nobiletin use in human therapy.This publication seems to be within the scope of journal. However it needs several corrections to be more acceptable for publication.

Because Molecules are an interdisciplinary journal, authors should consider explaining of all used abbreviations. Line 54 Please add examples of pregnancy-related diseases in development of which early appearance of apoptosis is important. Incorrect English abbreviation of milliliters – it should be mL in whole text. Lines 204, 217: it should be μL; Incorrect formula of carbon dioxide – it should be CO2 in line 178. In lines 176, 183, 195 it should be „density of 1×105 cells / mL” References should be formatted according to the instructions of the journal.

Author Response

Thanks for your careful read and thoughtful suggests on manuscript. The questions have been answered point-to-point.

Reviewer 3

Question one: Because Molecules are an interdisciplinary journal, authors should consider explaining of all used abbreviations.

Answer:

    Thank you for your suggestion.

Abbreviations

NOB

Nobiletin

BeWo cells

Human choriocarcinoma trophoblast cells

Cl-PARP

Cleaved PARP

DMSO

Dimethyl sulfoxide

FBS

Fetal bovine serum

PMSF

Phenylmethylsulfonyl fluoride

CU

Cellular uptake

PI

Propidium iodide

    The revisions were highlighted with red in the manuscript, line 309 , page 13.

Question two: Line 54 Please add examples of pregnancy-related diseases in development of which early appearance of apoptosis is important.

Answer:

    Thank you for your suggestion.

Before revision:

    However, the enhancement or early appearance of apoptosis may lead to pregnancy-related diseases.

After revision:

    However, the enhancement or early appearance of apoptosis may lead to pregnancy-related diseases, such as preeclampsia, stillbirth, and placental abruption (Kohan-Ghadr et al., 2019; Sagrillo-Fagundes, Salustiano, Ruano, Markus, & Vaillancourt, 2018; Yang, Xiao, Wang, Wang, & Wang, 2018).

    The revisions were highlighted with red in the manuscript, line from 55 to 56, page 2.

Question three: Incorrect English abbreviation of milliliters – it should be mL in whole text. Lines 204, 217: it should be μL; Incorrect formula of carbon dioxide – it should be CO2 in line 178. In lines 176, 183, 195 it should be „density of 1×105 cells / mL”

Answer:

    Thank you for your suggestion. The revisions were highlighted with red in the manuscript.

Question four: References should be formatted according to the instructions of the journal.

Answer:

Thank you for your suggestion. The revisions were highlighted with red in the manuscript.

References

Kohan-Ghadr, H. R., Kilburn, B. A., Kadam, L., Johnson, E., Kolb, B. L., Rodriguez-Kovacs, J., . . . Drewlo, S. (2019). Rosiglitazone augments antioxidant response in the human trophoblast and prevents apoptosisdagger. Biol Reprod, 100(2), 479-494. https://doi.org/10.1093/biolre/ioy186.

Sagrillo-Fagundes, L., Salustiano, E. M. A., Ruano, R., Markus, R. P., & Vaillancourt, C. (2018). Melatonin modulates autophagy and inflammation protecting human placental trophoblast from hypoxia/reoxygenation. Journal of Pineal Research, 65(4). https://doi.org/ARTN e12520 10.1111/jpi.12520.

Yang, A., Xiao, X. H., Wang, Z. L., Wang, Z. Y., & Wang, K. Y. (2018). T2-weighted balanced steady-state free procession MRI evaluated for diagnosing placental adhesion disorder in late pregnancy. Eur Radiol, 28(9), 3770-3778. https://doi.org/10.1007/s00330-018-5388-0.

Round 2

Reviewer 2 Report

Please accept the manuscript in the present form.

Reviewer 3 Report

The authors have carefully corrected the manuscript and the publication can be accepted in its current form.